# Design and Fabrication of an Automatic Fish Feeder Prototype Suits Tilapia Tanks

**Ahmed Mohamed El Shal** [1,*] **, Faisal Mohamed El Sheikh** [2] **and Atef Mohamed Elsbaay** [3]

1   Department of Agricultural Engineering, Zagazig University, Zagazig 44519, Egypt
2   Agricultural Research Centre, Agricultural Engineering Research Institute, El Dokki, Giza 12611, Egypt; f.elsheikh77@gmail.com
3   Department of Agricultural Engineering, Faculty of Agriculture, Kafrelsheikh University, Kafr El-sheikh 33516, Egypt; atef.ahmed@agr.kfs.edu.eg
*   Correspondence: amelshal@agri.zu.edu.eg; Tel.: +2-010-1444-9048

**Abstract:** The conventional methods of supplying feed to tilapia tanks are ineffective. It is better to find new a automatic feeder saving pellets from crushing and cohesion without hitting pellets during feeding at a predetermined interval of time and an accurate amount of food with a larger surface area covered by pellets. Developing-country fish farmers use manual feeding to be more cost-effective than with costly mechanized feeding, so this research aimed to design and construct an automatic fish feeder prototype to feed tilapia in a recirculation aquaculture system's tank. The performance of the prototype was studied after it was designed and installed. The dispensed feed operated by a DC motor located underneath the pellet hopper and the feed material was discharged into the tank through a gate in the bottom of the feeder. Three pellet sizes, three rotation speeds, and three feeder heights from the water's surface were used to test the automatic feeder's efficiency. The results showed that the optimal speed for the automatic feeder was 14 rpm with a height of 70 cm, resulting in a distribution width of 26.6 cm and a high automatic feeder efficiency of 99.9%. Furthermore, the feeder used very little electricity and saved time, cost, labor, energy, and pellets.

**Keywords:** aquaculture; automatic; electric; fish feeder; motor

## 1. Introduction

Feeding fish is labor-intensive and expensive. Feeding frequency is dependent on labor availability, farm size, as well as fish species and sizes. Generally, growth and feed conversion increase with feeding frequency. In intensive fish culture systems, fish may be fed as many as five times a day to maximize growth at optimum temperatures [1]. Since fish farmers must ensure that feeding regimes, feeding rates, ration sizes, and the period over which they are dispensed are adjusted to optimize consumption, growth, and conversion efficiencies, the expression of feeding behavior and the satiation time are both important. Furthermore, due to high feed costs and the possible negative effects of waste feed on water quality, wastage must be reduced [2,3]. The cost of raw material, as well as the equipment used in the production and processing of the feed, contribute to the high cost of aquaculture feed. The cost of aquaculture feeds can account for from 30 to 60% of total operating costs in aquaculture production [4].

The cost of feed is usually the greatest operating cost in aquaculture and may account for 40% or more of total operating costs in an eel culturing system [5,6]. The rise of automatic feeders has allowed farmers to increase the number of feedings without increasing the labor required [7]. This could be due to the fact that feed is not in the water for as long, resulting in less physical breakdown before consumption [8]. Feed stability requirements could be reduced if diets are attractive and consumed quickly. This may be relevant to feed management as feeds with lower stability may be formulated with lower-cost ingredients [9]. Another advantage of dynamic feed management is the possibility of lowering

waste from feed that is fed but not eaten. Overfeeding has also been shown to increase pond pollution and the expense of water quality management [10]. Most of the time overfeeding is related to the practice of feeding the fish until satiation is reached.

It is difficult to adjust feeding rations and avoid overfeeding without a way to monitor feed intake. The use of feeding trays to change feeding rates according to consumption has become commonplace [11]. The feeding rate can be set based on a percentage of the total body weight of fish being fed and adjusted for the size and the total number of fish and water temperature [12]. A suitable quantity is expressed as a percentage of one's average body weight. The percentage of body weight fed decreases as the weight of the fish increases. Fish are fed every 2–3 h and consume more food than their stomachs can handle. Because the excess food consumed goes through the stomach, it is termed waste. Fish that are fed every 4–5 h consume almost the same amount of food to replenish their stomachs. This means that, depending on the energy and composition of the diet, the optimum period between feedings is about 4–5 h [13].

When the temperature is high, metabolic processes increase, therefore, a high food demand is present; on the other hand, when the temperature is low, food demand decreases too [14,15]. A self-feeder's response times vary from 3 [16,17] to 15 s [3,18] and 1 min [19]. There are several automatic feeders which have been developed to fulfil certain objectives and requirements [20,21]. Increasing the number of daily feedings from two to six can allow for a higher daily feeding rate, which translates into higher growth and production [22]. Developing-country fish farmers find manual feeding to be more cost-effective than costly mechanized feeding. A simple human-powered fish feeding system was built in the study to solve the feeding problem in a freshwater aquaculture on a budget [23]. Each rearing tank measured 2.9 m in length, 1.4 m in width, and 1 m in depth. To keep the water volume at 3.2 $m^3$, the water level was kept at 80 cm deep. Per tank, 198 fish (60 fish per $m^3$) were stocked [24]. The basic information derived from the development of automatic feeder feeding strategies, tilapia weighing 185 g were fed 48 meals a day at 22 or 30 min intervals, with feeding rates of 2, 3, and 4% of live weight, in a study conducted from September to December. In one pond, eighteen 1 $m^3$ net cages with automatic feeders were spread. The mean values for dissolved oxygen, pH, and temperature for the experiment were 3.20 mg/L, 8.03, and 25.43 °C, respectively. The fish treated with 4% of their live weight at 30 min intervals between meals had the greatest average weight of 683.73 g. There was no difference in the viscerosomatic index, indicating that the ultimate carcass quality had not changed. These findings show that in the last phases of cage-raising Nile tilapia, a higher feeding rate combined with efficient feed management can be used without sacrificing end fish productive quality [25]. This system does not perform well with spread food or intensive feeding since it uses an infrared photoelectric sensor to start and stop feeding based on the gathering behavior under the feeder [20].

The problems of the old feeding system have been discussed, including inconsistent feed distribution, failure to add the correct amount of feed for fish, and harm caused by excess or lack of fish needs. Fish farming in tanks and ponds is a popular practice all over the world. The use of modern technologies is one way to ensure the availability and effectiveness of fish rearing in tanks on a long-term basis. Timer-controlled automatic feeders with a revolving plate are widely used in the indoor tilapia culturing industry. The aim of this research is to develop a simple prototype automatic fish feeder that can regulate single doses of fish pellets during each process.

## 2. Materials and Methods

The automatic fish feeder was installed in a private workshop in Mansoura, Daqahlia province, and tests were conducted in El Salhiya, El Sharkia province, Egypt.

### 2.1. Tank

One fiberglass tank with a surface area of 1 m$^2$ and a volume of 1 m$^3$ was used. The tank was filled with filtered pure water by 100 um pore size that ranged in temperature from 15 to 25 degrees Celsius.

### 2.2. Automatic Feeder Design, Fabrication, and Specification

The rate and type of particle flow in the feeder hopper will be affected by the hopper's configuration. The material flow function is one of the properties taken into account when designing the hopper. When some of it is drawn from the outlet gate, the mass movement of solids in the hopper, which was chosen as the motion of dry fish food, is uniform and in a steady state. This is the most important aspect of the design since it involves precise measurements to assess the size and opening of the outlet gate. The hopper was designed using the average bulk density of feed pellets of various sizes. We created this design to ensure that all pellets are drowned without clogging or cohesion pellets.

With a minimum height of 40 cm (H$_1$), a medium height of 55 cm (H$_2$), and a maximum height of 70 cm (H$_3$), the galvanized metal bar stand can be adjusted to the desired height between the bottom feeder hopper and the top of the water surface in the tank. The height can be easily modified to suit various tank heights. The model includes a cylinder container (Figure 1) that holds from 147.5 to 310 g of pellets and rotates step by step using an electric DC motor. It discharges from 29.5 to 62 g of fish food when the DC motor is tested for one separated by six rotations. As a result, the programing was set for 9, 11, and 14 rpm for testing purposes. As a result, it emitted from 29.5 to 62 g of fish pellets into the tank, which is adequate for the fishes' two-times-daily feeding rations.

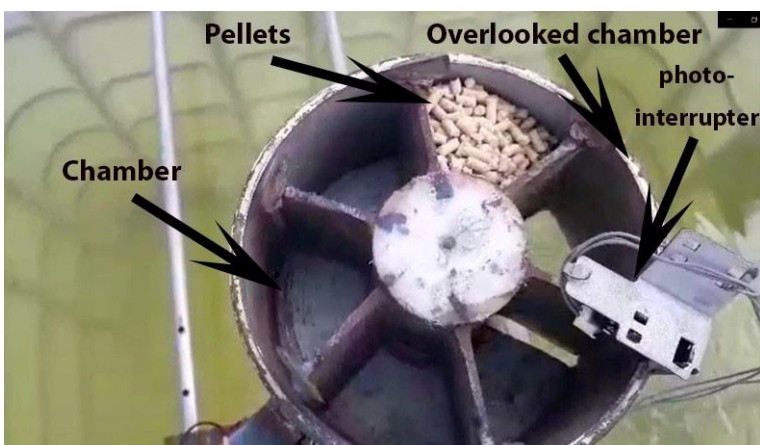

**Figure 1.** The fish feeder was connected to the power supply, and the photo interrupter was installed on top.

The cylinder container was made of galvanized metal and had a volume of 774.221 cm$^3$ with six chambers (one chamber was overlooked because it was over the gate position during the filling feeder), each of which had a volume of 129.037 cm$^3$ and rotated step by step with the aid of an electric DC motor operated by a control panel and LCD timing as shown in Figures 2 and 3. The rotation of the divided wheel with a metal piece fixed above the edge of each chamber passes through the photo interrupter, causing the electrical circuit to cut off and the rotation to stop at the next chamber. To empty one chamber, it discharges a maximum of 55, 62, and 29.5 g of fish feed during the gate in the bottom of the feeder, and the pellets chute discharges the feed material into the tank (P$_1$, P$_2$, and P$_3$, respectively). As a result, the software was set for 9, 11, and 14 rpm (S$_1$, S$_2$, and S$_3$, respectively) for testing purposes.

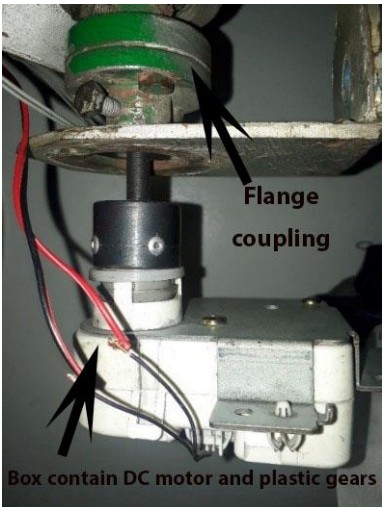
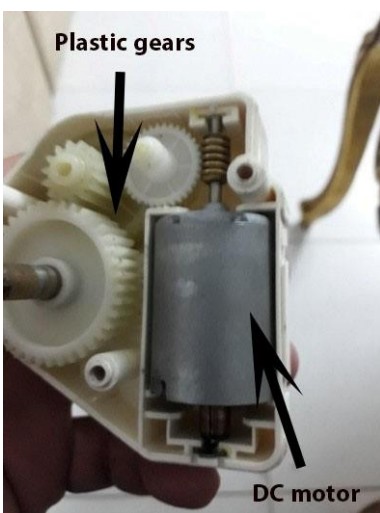

**Figure 2.** The motor connected to the feeder, and the DC motor transmission movement to small plastic gears.

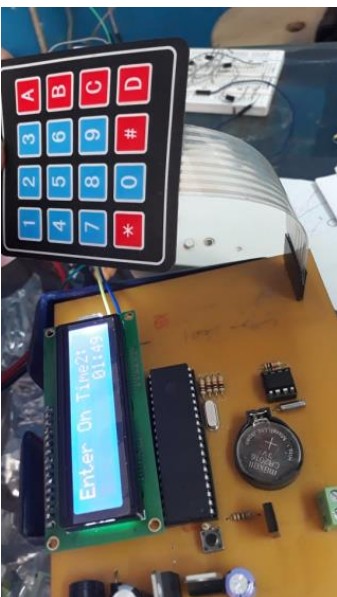

**Figure 3.** Control of the time on the LCD.

If the number of fish increases, the number of rotations of the rotating unit can be increased by programing the control unit as shown in Figures 4 and 5.

Aside from the feeding process, the automatic feeders allowed for an increase in daily frequency as well as feed input. This automated fish feeder was designed to meet specific goals and specifications, with added features such as a detachable hopper that can be adjusted to fit different hopper sizes depending on the needs of the user. It was also designed with adjustable height and speed to fit various fish tank sizes. Because of these characteristics, it is ideal for culturing tilapia and other fish species. When the next ration arrives, the feeder stops feeding, reducing the possibility of water contamination and other issues. Therefore, no residue contaminates the water, and the dosage size and application timing are set.

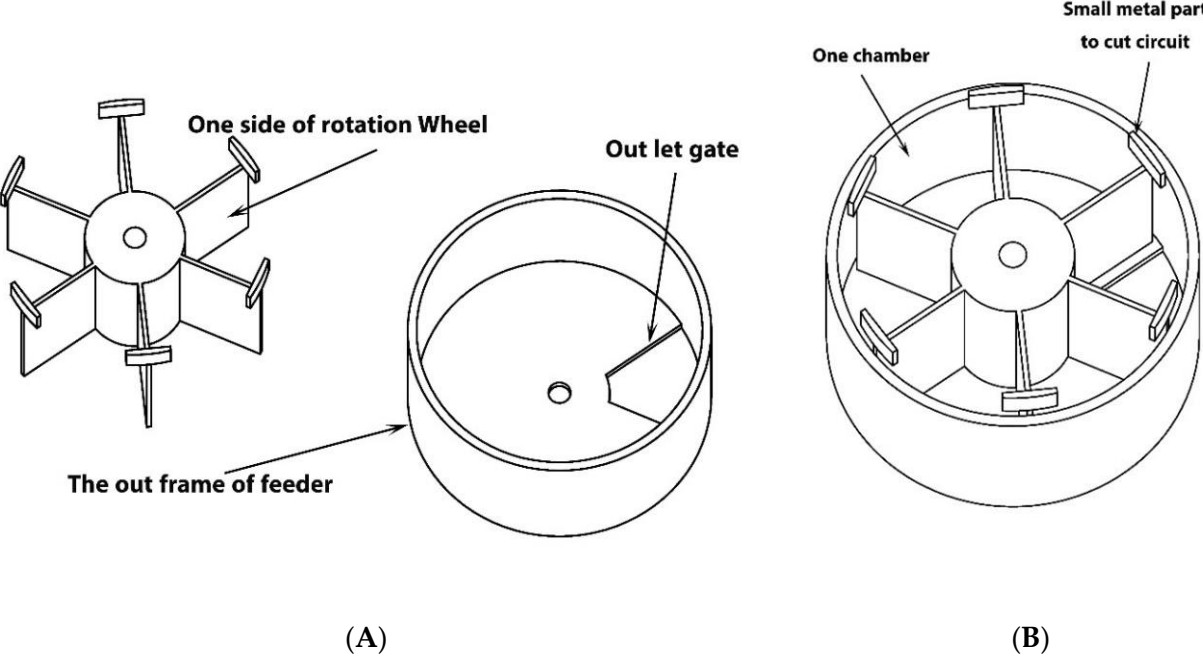

(**A**)                                                  (**B**)

**Figure 4.** The components of the feeder before (**A**) and after (**B**) assembly.

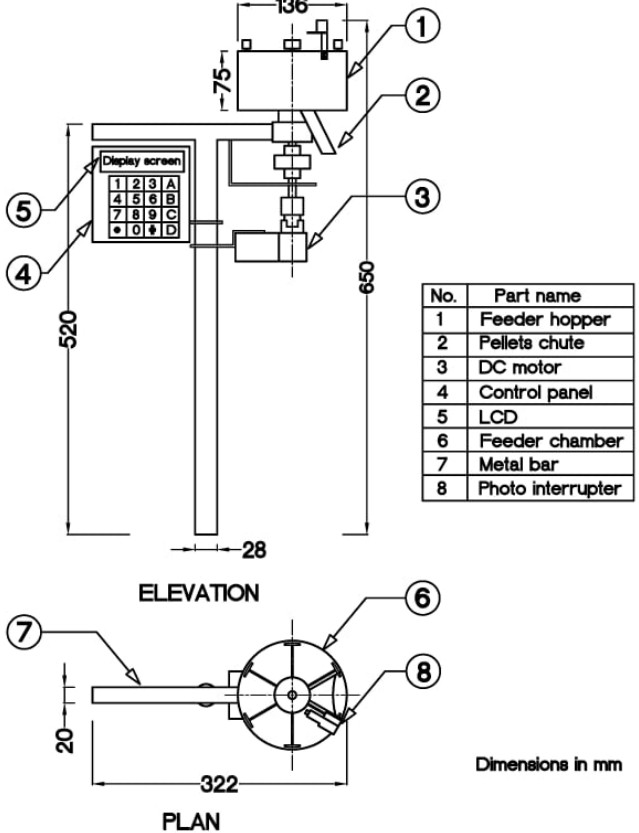

| No. | Part name |
|-----|-----------|
| 1 | Feeder hopper |
| 2 | Pellets chute |
| 3 | DC motor |
| 4 | Control panel |
| 5 | LCD |
| 6 | Feeder chamber |
| 7 | Metal bar |
| 8 | Photo interrupter |

**Figure 5.** Elevation and plan view of all the components of the automatic feeder.

*2.3. Fish Pellets*

Table 1 shows three types of industrial fish feed pellets: tiny floating pellets, medium submersible pellets, and large floating pellets. All the used feed were cylindrical, the height and diameter of the base for each of them were as follows (large, medium,

and small) (5 mm, 3.5 mm), (3.75 mm, 3 mm) and (3 mm, 2 mm), respectively (see also Figure 6). The entire amount required for the experiment had been purchased ahead of time. A representative sample was taken from each bag, the samples were then combined and analyzed.

**Table 1.** Three-size fish pellets' moisture contents and bulk density.

| Fish Pellets Characteristic | Moisture Content (%) | Bulk Density (g/cm$^3$) |
|---|---|---|
| Small floating pellets (P1) | $9.36 \pm 0.11$ | $0.427 \pm 0.03$ |
| Medium submersible pellets (P2) | $8.46 \pm 0.13$ | $0.481 \pm 0.01$ |
| Large floating pellets (P3) | $8.52 \pm 0.10$ | $0.229 \pm 0.01$ |

Each value was mean $\pm$ SD.

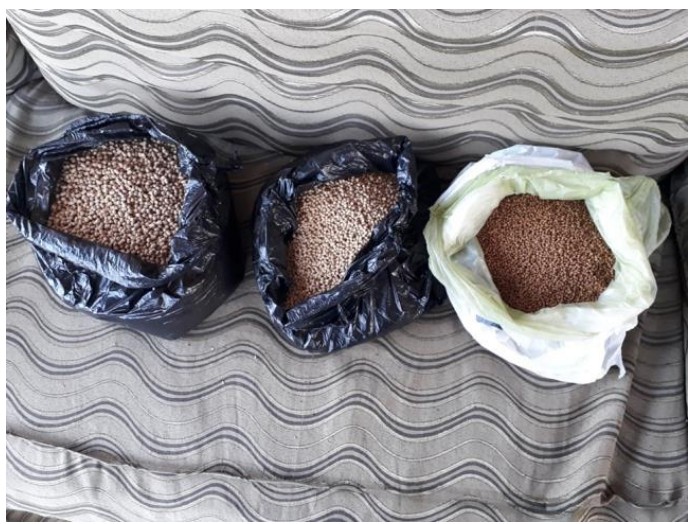

**Figure 6.** Fish pellets from large to small size (from left to right).

### 2.4. Experimental Conditions

Three user-adjustable speeds with default values were included in the control plan for the automatic fish feeder unit. Three pellet sizes $P_1$, $P_2$, and $P_3$, feeding twice a day, three feed rations $W_1 = 55$ g, $W_2 = 62$ g, $W_3 = 29.5$ g, and feeder hopper height 40, 55, and 70 cm equal to $H_1$, $H_2$, and $H_3$, respectively, as shown in Figure 7.

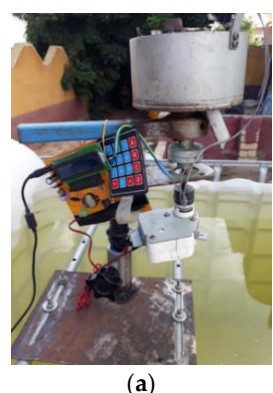 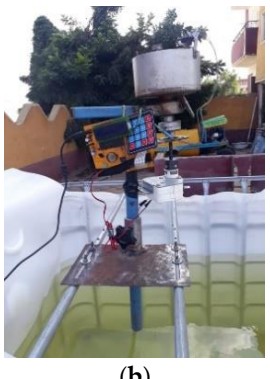 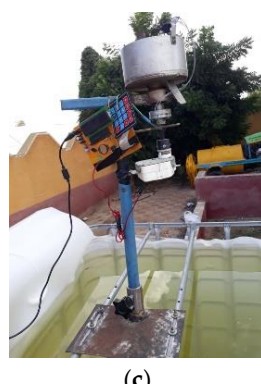

| (a) | (b) | (c) |
|---|---|---|

**Figure 7.** Lowest height (**a**), medium height (**b**) and highest height (**c**).

### 2.5. Instruments

A scale balance (OHAUS-USA) model made in China, Guangzhou, was used for massing the fish pellet samples. A laser tachometer was used for measuring the rotating speed of the motor shaft. To measure the fish pellets diameter, a mechanical Vernier caliper

with an accuracy of 0.01 mm was used. Moisture content was determined based on [26]. The moisture content of the feed was determined using a drying oven at 105 °C for 24 h, and the feed was weighed before and after drying. To measure hardness, a Digital Force Gauge model FGN-50 with an accuracy of 0.2% of maximum load +1/2 digit at 23 °C, a weight of approximately 450 g, a resolution of 0.1 N, and a measuring range of 500 N, and powered by a NiCd battery and an AC adapter (DC 9-volt 200 mA), was used [27]. The standard for the durability of pellets and crumbles is the Pellet Durability Test.

The dispensing distance was measured with a wooden meter ruler.

### 2.6. Measurements

The automatic fish feeder's productivity was calculated from the following equation:

$$P_f = M_d / T \tag{1}$$

where $P_f$ is fish feeder productivity, g/h; $M_d$ is mass of ration, g; T is consumed time, h.

Automatic feeder efficiency was calculated from the following equation:

$$A_{FE} = (Q_d / Q_i) \times 100 \tag{2}$$

where $A_{FE}$ is automatic feeder efficiency, %; $Q_d$ is the quantity of the discharge, g; $Q_i$ is the quantity of the feed input into the feeder, g.

The following formula was used to estimate the power consumed:

$$P = I \times V \tag{3}$$

where P is required power, W; I is current intensity, A; V is voltage, V.

The hourly cost of the automatic feeder was determined by [28]:

$$C = p/h \ (1/a + i/2 + t + r) + (P \times e) + m/144 \tag{4}$$

where C is the cost for working during one hour, USD/h; P is the price of the automatic feeder, USD; h is working hours per one year, h/y; a is the life expectancy of the automatic feeder, y; i is the rate of interest for one year, %; t is the ratio of taxes overheads, %; r is repairs and maintenance ratio, %; P is the consumed power, kW; e is the price of kilowatt per hour, USD/kW h; m is the monthly salary for an operator, USD; 144 is the average number of working hours per month, h.

$$D = \frac{WPAT}{WPBT} \times 100 \tag{5}$$

where D is durability of pellets, %; WPAT is the weight of pellets or crumbles after tumbling g; WPBT is the weight of pellets or crumbles before tumbling (500 g).

### 3. Results and Discussion

Automatic feeders still stand out as the best option for feeding fish to replace the traditional methods. It is better to find new automatic feeders saving pellets from crushing and cohesion without hitting pellets during feeding at a predetermined interval of time and an accurate amount of food, with a larger surface area covered by pellets. There are still challenges for developing-country fish farmers to find manual feeding to be more cost-effective than costly mechanized feeding.

### 3.1. Productivity

The automatic fish feeder was operated by a digital timer and could feed the fish according to a predetermined time schedule without the need for an operator, at an average feeding rate ration of 48.8 g for a one-time feeding.

A digital timer controlled the aquaculture, allowing the owner to change the cycle time and dispensing time as required. More significantly, the timing can be configured to maintain a stable feeding schedule. The feeding mechanism can be adjusted quickly and in a variety of ways. The height of the feeder, for example, can be adjusted. The device's height above the water surface was for increasing the feed area distribution. In addition, the hopper size can be adjusted to accommodate a different feed volume. Furthermore, the opening angle may be modified to suit various tank or pond sizes [1].

Table 2 shows the durability of three different pellet sizes: small floating pellets, medium submersible pellets, and large floating pellets, based on examination of three replicates. $P_1$ had the maximum durability of 99.29%, $P_3$ had the lowest durability of 92.81%, and $P_2$ had the medium durability of 98.92%. It is possible that bigger pellets were less robust and more prone to becoming brittle or crashing.

**Table 2.** Durability of fish pellets of three sizes.

| Fish Pellets Characteristic | Durability (%) |
|---|---|
| $P_1$ | 99.29 |
| $P_2$ | 98.92 |
| $P_3$ | 92.81 |

In terms of growth and production, automated feedback regulation of feed inputs outperforms hand feeding twice a day or the use of simple automatic feeding systems with higher feed inputs for shrimp [22].

The overall hardness was 41.1 N with medium submersible pellets, and the minimum hardness was 2.3 N with large floating pellets, according to the hardness average for three sizes. The average hardness of small floating pellets, medium submersible pellets, and large floating pellets was around 21.22, 29.15, and 6.19 N, respectively, as shown in Table 3.

**Table 3.** The hardness of small floating pellets, medium submersible pellets, and large floating pellets.

|  | $P_1$, N | $P_2$, N | $P_3$, N |
|---|---|---|---|
|  | 22.4 | 37.2 | 6.6 |
|  | 17.2 | 26.4 | 10.1 |
|  | 28.3 | 41.1 | 5.9 |
|  | 13.4 | 31.2 | 2.3 |
|  | 21.6 | 23.8 | 3.5 |
|  | 26.9 | 24.1 | 6.7 |
|  | 15.5 | 29.3 | 10.1 |
|  | 28.9 | 21.8 | 7.1 |
|  | 17.9 | 27.2 | 5.6 |
|  | 20.1 | 29.4 | 4 |
| Avg. | 21.22 | 29.15 | 6.19 |
| Max. | 28.90 | 41.10 | 10.10 |
| Min. | 13.40 | 21.80 | 2.30 |
| Sd. | 5.15 | 5.76 | 2.44 |
| Cv. | 0.24 | 0.20 | 0.39 |

## 3.2. Efficiency

The automatic feeder's performance was found to be optimal as compared to that of other feeders because the design drowned all pellets down without clogging or cohesion pellets in this analysis. The automatic feeder is adaptable and can be reassembled to suit any hopper size or height requirement. The automatic feeder's efficiency was 99.9% for all tests under three heights, three rotation speeds, and three pellet sizes, indicating that substantial dispensing distance, accurate time, and pellet saving from crushing and cohesion were all achieved. This saves time, effort, pellets, and money. Improving feed

efficiency is key to reducing production costs in aquaculture and to achieving sustainability for the aquaculture industry [29].

### 3.3. Spread

The findings are supported by the results obtained at various operating speeds (from 9 to 14 rpm). The pellets would be dispensed even further when the hopper is fixed at a higher level. For small, medium, and large pellets, the dispensing widths increased with increasing motor speed and hopper height, as shown in Figure 8. The distribution width of pellet sizes being dispensed increased as the hopper position was increased. Several experiments were also carried out to assess the total amount of fish pellets dispensed at various intervals. Various quantities of fish pellets used for this (55, 62, and 29.5 g) were separately put in the hopper.

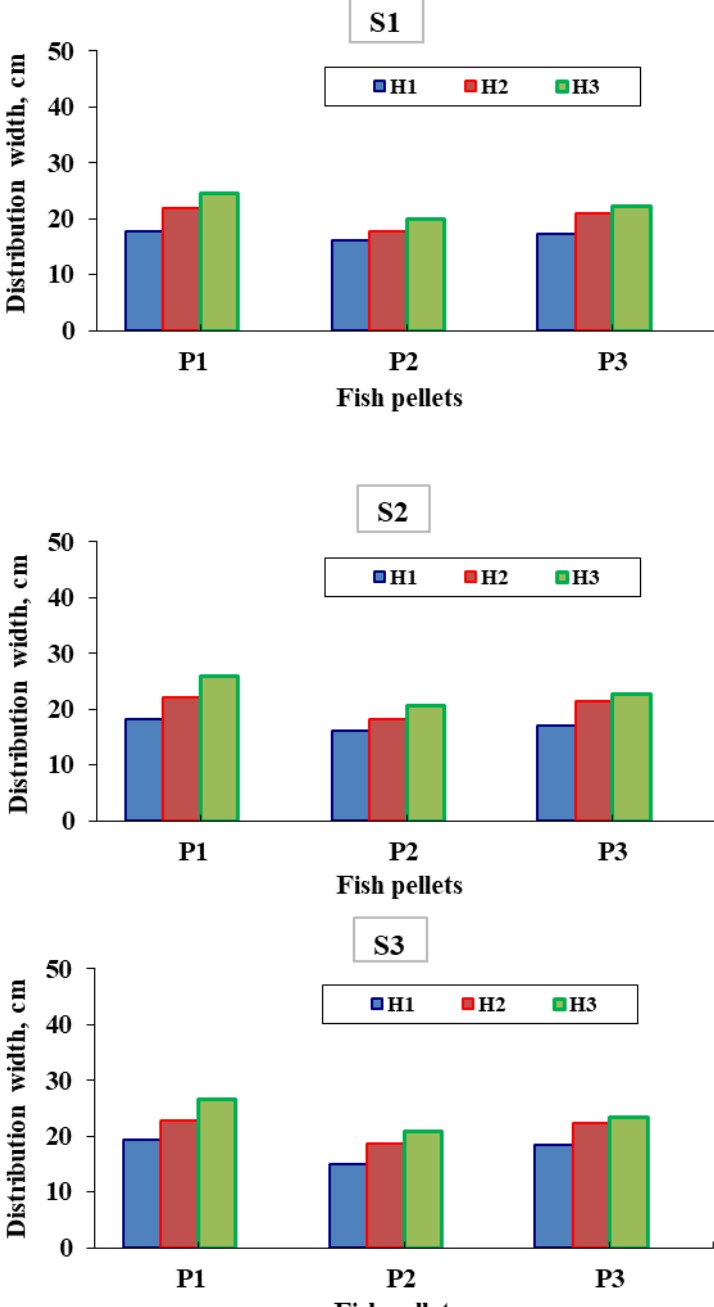

**Figure 8.** Effect of rotation speed and feeder height for different pellet sizes on distribution width.

The dispensing rate for small floating pellets was about 275 g, 310 g for medium pellets, and 147.5 g for large floating pellets. A moisture content analysis of the pellets in the hopper was also performed. The moisture content of the pellets remained constant during the test, suggesting that the feeder was moisture-resistant.

The small floating pellets had the maximum distribution average width of 26.6 cm at 14 rpm rotation and feeder height of 70 cm, while the medium submersible pellets had the lowest distribution average width of 14.9 cm at 14 rpm rotation and feeder height of 40 cm. The fact that the optimal distribution was at a quick pace for the highest high for small pellets was clarified by these findings, which could be due to the small volume of pellets. When all the fish in the tank are fed in the same way and have equal access to food, competition is minimized [30].

Similarly, when the position of the hopper is raised, the dispense width of the small and large pellets will also increase [1].

### 3.4. Power

The estimated and determined power during the measured voltage and current results showed that the motor power increased as the revolutions increased. This may be due to the fact that, as seen in Table 4, power increased linearly with increasing revolution speed.

**Table 4.** Power consumed at different speeds.

| Speed of Rotating, rpm | Power, W |
|---|---|
| S1 | 0.91 |
| S2 | 1.37 |
| S3 | 1.84 |

### 3.5. Operational Cost

The cost of feed is normally the highest operating cost in aquaculture. Overfeeding results in leftover feed, which causes poor water quality, stress for the fish, and additional load on the mechanical filters, biofilters, and aeration equipment, not to mention the additional expense. It is just as important to control feeding as it is to plan a diet, and knowing when to stop feeding is crucial.

It is claimed that the cost of the automatic feeder with one operator can be ignored when measuring cost because of the small use of electric power of a few watts and the cost of about 1.7 USD/h for the first feed for half day and on their day, they did not require a worker for two feeds, implying that there are three rations for two days. One worker would cost 1.6 USD/h if we used manual feeding and put in a lot of effort and went to the tank farm twice a day. These days, workers are not required and only cost less than 0.2 USD/h because it was not taken into account that there is a need to be at a fish farm and that a DC motor only works for a few seconds rather than an hour. These results are consistent with the finding of [31], who claimed that the AQ1 feeder could be a feasible, low-labor, low-cost option for the shrimp commercial industry. However, such technologies must be matched to the production system's capacity to process nutrients. Automatic and demand feeders save time, labor, and resources, but they do so at the cost of the diligence needed for handfeeding [32]. Tilapias are also one of the most simple and lucrative fish to raise. This is due to their omnivorous diet, reproductive mode, disease tolerance, and rapid growth [33]. They grow best when fed two or three times a day, although adequate growth can be obtained with a single daily feeding [34]. Fish feeding accounts for about 40% of overall production costs in intensive aquaculture systems. Food surpluses result in unconsumed food deposits on the tank floor, and their decomposition consumes oxygen and produces ammonia nitrogen. In indoor recirculating eel culturing, timer-controlled automatic feeders with a revolving plate and scrubber are commonly used [20]. Feed costs account for 30–70% of total production costs in aquaculture [29].

## 4. Conclusions

This study gives a practical model for feeding fish, allowing a fish owner to relax knowing that the fish would be fed at the appropriate times. This is useful when the owner is away or too busy to keep up with the regular feeding routine. A simple prototype automatic fish feeder that could adjust single doses of fish pellets during each step was successfully developed. An auto feeder is a fully automated system that allows one to select the frequency and quantity of feed using a digital control system that is controlled by a microcontroller built into the feeder. It is simple to operate with just a touch of a finger, and the feeding system works at all hours of the day and night. The treatments employed indicated that the feeder optimal speed was 14 rpm with a height of 70 cm, resulting in a distribution average width of 26.6 cm and a high automatic feeder efficiency of 99.9% for all tests. Proper feed distribution, feeding timing, and higher feeding frequency limit pellet exposure time in the water, reducing the negative consequences of water-soluble pellet leaching.

**Author Contributions:** Conceptualization, A.M.E.S. and F.M.E.S.; methodology, A.M.E.S. and F.M.E.S.; investigation, A.M.E.S. and F.M.E.S.; writing—original draft preparation, A.M.E.S., F.M.E.S. and A.M.E.; primitive and final field experiments, A.M.E.S., F.M.E.S. and A.M.E.; writing—review and editing, A.M.E.S., F.M.E.S. and A.M.E.; All authors have read and agreed to the published version of the manuscript.

**Funding:** This research received no external funding.

**Acknowledgments:** The authors thank their family members.

**Conflicts of Interest:** The authors declare no conflict of interests.

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
