# Peer review of "Design and Fabrication of an Automatic Fish Feeder Prototype Suits Tilapia Tanks"

_fishes, doi:10.3390/fishes6040074_

Round 1
Reviewer 1 Report
A very interesting article, fitting with the domain of a scientific journal.
I read your article with great interest, especially in terms of the design solution, which is innovative in nature. Interesting idea, well described, with good methodology and research but with a very poor ending that needs to be rewritten.
Substantive Evaluation:
- Purpose and scope consistent with the title of the article,
- methodology proves correct,
- research conducted in a manner consistent with applied science medolology,
- description of results clear and understandable,
- Conclusions - incomprehensible, moreover they are not conclusions but a summary, which is a repetition of what the authors have previously studied. This chapter should be completely rewritten, writing 2-3 conclusions that respond to the stated purpose of the paper.
- The literature is relevant to the topic, the references are correct.
I believe that the work is suitable for printing after correction of the conclusions.
Author Response
Thank you very much

Reviewer 2 Report
Dear authors, I attached a word text with my comments to your manuscript.
I liked to read the process that you follow to design and test a prototype for aquaculture purposes.
Manuscript Number: FISHES-1447393
Title: Design and Fabrication of an Automatic Fish Feeder Suits Tilapia Fry Tanks
Ahmed Mohamed El Shal and Faisal Mohamed El Sheikh
About the overall paper:
The paper performs a study to develop a mechanical feeder prototype to feed tilapia fry. Then, the prototype was tested in a small vessel to gather experimental data regarding the capabilities and the efficiency to handle the feed to the rearing tank.
There are many issues in this manuscript, which I am going to point out in detail below. I would like to read your improved manuscript again, before recommending the editor the approval for its publication.
The paper title will be more adequate for the research attempted to be published by adding the word “prototype”.
Final disposition:
I would approve this manuscript to be published after the authors make substantial changes as addressed below.
General Comments:
The cited literature throughout your manuscript must be ordered by year, from oldest to most recent (i.e., lines 91, 93, 107, 108).
The introduction should include a little information about the volume and surface area of typical tanks and ponds used for commercial fry farming. Following with a description on how the feed is provided for the fry reared in those tanks and ponds.
The discussion section should address a comprehensive comparison between the current technology to feed fry and the proposed prototype.
Overall the manuscript should address the fry rearing in tanks and ponds, and problems associated for their conventional feeding. However, along the text you always are referring to fish. This confuse the reader, since is unclear is your solution is for early fish feeding stages or for the whole growout fish commercial production.
Abstract
The abstract must be rewritten to state what you really performed in your study, developing a prototype to feed tilapia fry. Therefore remove any ideas that could confuse the reader that you might be designing something to feed fish larger than fry.
Line 21. List your keywords alphabetically.
Introduction:
Focus your introduction in the feeding of fry. You are focusing your solution to fry and not for larger fish stages development.
Line 35 to 36. It is ok to mention the cost to other fish (i.e. eels), but you cannot miss referring to the cost related to early feeding stages of tilapia.
Line 38 to 39. Rewrite the sentence “This could be a result of the less physical breakdown of feed before consumption as it is not in the water for as long”.
Line 43 to 44. You should also comment that most of the time overfeeding is related to the practice of feeding the fish until satiation is reached.
Line 52 to 53. Rewrite this sentence “The excess food consumed is considered wasted since it passes into the stomach”. It is awkward to read that food that goes into the stomach is considered as a wasted food.
Line 53 to 54. Erase the sentence “As a result…”, since there is no relation between feeding the fish and the manufacturing cost of the fish feed.
Line 60. “… to 15 seconds [18,3]. and …”. Erase the period and write a comma “…[18,3], and ...”
Line 67. … practice …
Line 70. I suggest to rewrite your goal as “The aim of this research is to develop a simple prototype automatic fish feeder that can regulate single doses of fish pellets during each process.”. What do you mean with “each process”?
Lines 72 to 80. These prototype design restrictions should go in the 2.2. methods section. These restrictions define the operational characteristics to which the prototype must conform.
Materials and Methods.
Line 85. …fiberglass…
Line 86. State the degree of filtration (i.e. 100 um, 50 um, 10 um, ??)
Line 103. Which software are you referring to?
Line 109. Is it necessary so many significant numbers?
Line 110. It should be “five” instead of “six”, since the number 645 cm3 was obtained from 129 cm3 x 5
Line 133. Describe the size in mm for each of the pellets tested in your device. In line 154 you describe the use of a Vernier caliper to measure the fish pellets, but none what shown in the results or methods section. I am assuming that you are testing pellets used regularly to feed tilapia fry.
Line 155. Please describe the method to determine moisture content.
Line 156 to 158. Please check the grammar of the sentence
Line 159 to 160. Please check the grammar of the sentence
Line 166. What is really measuring this automatic feeder efficiency? How do you trap the discharged quantity of feed?
Line 171 and Line 174. Why do you have different letters for the same item? P is the power consumed in Equation 3, and W is consumed power in Equation 4.. You should use the same letter for the same expression.
Line 174. Where did you get this equation? Please cite the reference.
Line 181. Durability of what?
Line 182 to 183. Why the weight of pellets will change before and after tumbling?
What statistics do you apply to determine if there are differences between the types of durability?
Results and Discussion
Line 192. I am confused with your statement that “The height of the feeder is to match various tank heights…”. I thought that the device height above the water surface was for increasing the feed area distribution, if so then state it that way. In addition, it is needed in your article a table disclosing typical tilapia fry rearing tanks describing height, surface area, diameter, fry rearing density.
Line 197 to 199. How many durability test did you perform per each pellet size?
Lines 202 to 203. What you are claiming here is either not true or was not proven in your experiment. Therefore, I suggest that you rephrase your writing.
Line 205 to 209. Are the pellet hardness significantly different among the pellets sizes?
Line 213. “…as compared to other feeders..”. I do not see any comparison with other feeders. Can you include a table with other manufactured feeders efficiencies with their respective references?
Line 218. How did you measure the dispensing distance?. It was not described in your methodology.
Line 225. “…level (Fig. 8).” Erase from the sentence “…, as seen in the diagram.”
Line 238. Did you make a moisture resistant feeder? Can you explain it in the Method section?
Line 248. Does the required power increase with size and weight of the pellet?
Line 254 to 256. This idea have to be further developed to provide an interesting discussion. Otherwise, we are not learning too much from this sentence.
Lines 266 to 270. These are very awkward sentences. Please rewrite so they can be understood.
Line 276. “…rapid development.”, change it to “…rapid growth.”
Line 278. This statement is not true. Fish have quite good conversion rate, and they are reported to be between 0.8 to 1.5. Prove me wrong with a better review of conversion rate´s for tilapia.
Line 281. You just cited research with eels. What about other fishes such as turbot, yellowtail kingfish, Atlantic salmon, rainbow trout, and so on. There are many fish reared in land-based systems that are fed with automatics feeders. I suggest you include in your discussion a description of automatic technologies being using in commercial aquaculture worldwide.
Line 281. “re-circulating”, better “recirculating”
Line 284. This single sentence is awkward.
Conclusion
Lines 286 to 291. This is not a conclusion. I suggest move them to the initial section of your Results and Discussion.
Line 291. “This study provides a non-convenient way of feeding fish…”. I am confused. Did not you just say that your mechanical feeder is a very convenient option for feeding fish? So, why now you say that it is not convenient?
This section has to be rewritten since you are mixing results, discussion and conclusion. You should only present here your conclusion, which must be based on your proposed main goal.
Figures
Fig 1. It is difficult to identify the parts. It would help if you add some arrows indicating the main parts.
Fig. 2. Same comments as for Fig. 1
Line 131 and Line 139 You have two figure 5
Line 139. Fig. 5. It seems to me that the big size pellets are at the left side of the picture. And the smallest pellets size at the right side.
Tables
Table 1. Can you write the standard deviation of each data?.
References
You need to write scientific names in italic in your references.
Author Response
Thank you very much

Round 2
Reviewer 1 Report
I accept the corrections and recommend them for printing
Author Response
Thank you very much.
Reviewer 2 Report
Dear authors, at this time you have improved your article. However, I still have a few observations.

Author Response
Thank you very much.
